# LoRA-Pro: Are Low-Rank Adapters Properly Optimized?

**Zhengbo Wang**[1,2]    **Jian Liang**[2,3*]  **Ran He**[2,3]    **Zilei Wang**[1]    **Tieniu Tan**[2,4]

[1] University of Science and Technology of China
[2] NLPR & MAIS, Institute of Automation, Chinese Academy of Sciences
[3] School of Artificial Intelligence, University of Chinese Academy of Sciences [4] Nanjing University
`zhengbowang@mail.ustc.edu.cn`, `liangjian92@gmail.com`

## Abstract

Low-rank adaptation, also known as LoRA, has emerged as a prominent method for parameter-efficient fine-tuning of foundation models. Despite its computational efficiency, LoRA still yields inferior performance compared to full fine-tuning. In this paper, we first uncover a fundamental connection between the optimization processes of LoRA and full fine-tuning: using LoRA for optimization is mathematically equivalent to full fine-tuning using a low-rank gradient for parameter updates. And this low-rank gradient can be expressed in terms of the gradients of the two low-rank matrices in LoRA. Leveraging this insight, we introduce LoRA-Pro, a method that enhances LoRA's performance by strategically adjusting the gradients of these low-rank matrices. This adjustment allows the low-rank gradient to more accurately approximate the full fine-tuning gradient, thereby narrowing the performance gap between LoRA and full fine-tuning. Furthermore, we theoretically derive the optimal solutions for adjusting the gradients of the low-rank matrices, applying them during fine-tuning in LoRA-Pro. We conduct extensive experiments across natural language understanding, dialogue generation, mathematical reasoning, code generation, and image classification tasks, demonstrating that LoRA-Pro substantially improves LoRA's performance, effectively narrowing the gap with full fine-tuning. Our code is publicly available at `https://github.com/mrflogs/LoRA-Pro`.

## 1 Introduction

Foundational models (Radford et al., 2021; Brown et al., 2020; Achiam et al., 2023; Kirillov et al., 2023; Rombach et al., 2022; Touvron et al., 2023) have become the cornerstone of modern deep learning. By undergoing pre-training on massive datasets, these models typically exhibit excellent generalization and versatility. Remarkably, some foundation models even demonstrate emergent properties (Hoffmann et al., 2022; Kaplan et al., 2020). Due to these advantages, foundational models have been widely applied to various downstream applications.

Nevertheless, it still requires additional fine-tuning when applied to downstream tasks, where the huge parameter size of foundation models result in high cost in this stage. To address this issue, recent research has focused on parameter-efficient fine-tuning (PEFT) methods (Hu et al., 2022; Houlsby et al., 2019; Lester et al., 2021). PEFT methods reduce the fine-tuning cost by keeping the foundation models frozen and only fine-tuning small, additional lightweight adapters. With the majority of parameters frozen, PEFT enables faster fine-tuning and requires fewer resources.

Low-rank adaptation (Hu et al., 2022), also known as LoRA, is one of the most famous PEFT methods, which has been widely adopted across various domains. Inspired by previous works (Aghajanyan et al., 2021; Li et al., 2018), LoRA hypothesizes that the changes in weights during model adaptation exhibit a low-rank structure. To capture this, LoRA re-parameterizes these changes by expressing them as the product of two low-rank matrices: $W = W_0 + \Delta W \approx W_0 + sBA$, where $s$

---

*Corresponding author.

is a scaling factor, and $A \in \mathbb{R}^{r \times n}$ and $B \in \mathbb{R}^{m \times r}$ are low-rank matrices with rank $r \ll \min(m, n)$. LoRA reduces the number of trainable parameters from $m \times n$ to $r \times (m + n)$, thereby decreasing the cost of fine-tuning. However, despite its efficiency, LoRA's fine-tuning performance often falls short compared to full fine-tuning (Hu et al., 2022; Liu et al., 2024; Ding et al., 2023).

In this paper, we propose a novel PEFT method, LoRA-Pro, aimed at bridging the gap between LoRA and full fine-tuning. To begin with, we uncover a crucial connection between the optimization processes of LoRA and full fine-tuning: using LoRA for optimization is equivalent to full fine-tuning using a low-rank gradient for parameter updates. In LoRA, we discover that the change in weight $W$ is connected to the changes in matrices $A$ and $B$, expressed as $\mathrm{d}W = \frac{\partial W}{\partial A}\mathrm{d}A + \frac{\partial W}{\partial B}\mathrm{d}B$. This relationship implies that updating matrices $A$ and $B$ with gradients $g^A$ and $g^B$ is equivalent to updating $W$ with a low-rank equivalent gradient $\tilde{g}$ in full fine-tuning, where:

$$\tilde{g} = \frac{\partial W}{\partial A}g^A + \frac{\partial W}{\partial B}g^B = sBg^A + sg^B A. \tag{1}$$

Leveraging this insight, our goal is to bridge LoRA's gap with full fine-tuning by minimizing the discrepancy between the low-rank equivalent gradient $\tilde{g}$ and the full fine-tuning gradient $g$, by adjusting the gradients of matrices $A$ and $B$, i.e., $\min_{g^A, g^B} \|\tilde{g} - g\|_F^2$. Furthermore, we theoretically demonstrate that this optimization problem admits an optimal closed-form solution, as shown in Theorem 2.1. **Notably, the optimal gradients for the low-rank matrices do not explicitly depend on the full fine-tuning gradient.**

Our main contributions are summarized as follows:

- We first uncover a crucial connection between LoRA and full fine-tuning in optimization process: optimizing with LoRA is mathematically equivalent to full fine-tuning using a low-rank gradient for updating.
- We propose a novel PEFT method called LoRA-Pro. Our approach minimizes the distance between the true gradient and the low-rank gradient by adjusting the gradients of matrices A and B. We theoretically prove the optimal gradients and optimize using these gradients.
- Extensive experiments across tasks in natural language understanding, dialogue generation, mathematical reasoning, code generation, and image classification demonstrate the effectiveness of our method.

## 2 METHOD

In this section, we begin by revisiting LoRA (Hu et al., 2022) in Section 2.1. Following this, we conduct a comparison between LoRA and full fine-tuning, and point out their connection in the optimization process in Section 2.2. Finally, in Section 2.3, we introduce LoRA-Pro as a solution to bridge the gap between LoRA and full fine-tuning.

### 2.1 REVISITING LOW-RANK ADAPTATION

First of all, let's dive back into Low-Rank Adaptation (LoRA) (Hu et al., 2022). LoRA's core idea revolves around recognizing the low-rank structure of the change matrix $\Delta W$ in the standard fine-tuning process. This insight allows LoRA (Hu et al., 2022) to re-parameterize the change matrix into the product of two low-rank matrices,

$$W = W_0 + \Delta W \approx W_0 + sBA. \tag{2}$$

Here, $W_0 \in \mathbb{R}^{m \times n}$ represents the pre-trained weight matrix, $B \in \mathbb{R}^{m \times r}$ and $A \in \mathbb{R}^{r \times n}$ are the low-rank matrices, and $s$ is a scaling factor. For LoRA (Hu et al., 2022), $s = \frac{\alpha}{r}$, while for rsLoRA (Kalajdzievski, 2023), $s = \frac{\alpha}{\sqrt{r}}$. Here, $\alpha$ is the hyper-parameter and $r \ll min(m, n)$ denotes the rank. Consequently, LoRA significantly reduces the number of fine-tuning parameters from $m \times n$ to $r \times (m + n)$, thereby decreasing the computational cost of fine-tuning.

### 2.2 LORA V.S. FULL FINE-TUNING

Despite its widespread applications across various domains, LoRA's performance still falls short when compared to full fine-tuning. In this part, we compare LoRA and full fine-tuning in the op-

timization process. Then, we demonstrate that optimizing using LoRA is equivalent to using a low-rank gradient in full fine-tuning for updating the parameters.

**Full fine-tuning.** In full fine-tuning, we utilize differential to analyze the relationship between changes in the loss and changes in the weights:

$$dL = \langle \frac{\partial L}{\partial W}, dW \rangle_F, \tag{3}$$

where $dL$ and $dW$ denotes the changes of the parameter $W$ and the loss $L$, and $\langle \cdot, \cdot \rangle_F$ is the Frobenius inner product. To minimize the loss function, we typically set $dW = -\frac{\partial L}{\partial W} \triangleq -g$ (omitting the learning rate for simplicity), which results in $dL = -\|\frac{\partial L}{\partial W}\|_F^2 \le 0$.

**Low-rank adaptation.** In LoRA optimization, given that $W = W_0 + sBA$, we compute the differential using the chain rule:

$$\begin{aligned} dL &= \langle \frac{\partial L}{\partial W}, dW \rangle_F \\ &= \langle \frac{\partial L}{\partial W}, \frac{\partial W}{\partial A}^T dA + \frac{\partial W}{\partial B}^T dB \rangle_F \\ &= \langle \frac{\partial L}{\partial W} \frac{\partial W}{\partial A}, dA \rangle_F + \langle \frac{\partial L}{\partial W} \frac{\partial W}{\partial B}, dB \rangle_F \\ &= \langle \frac{\partial L}{\partial A}, dA \rangle_F + \langle \frac{\partial L}{\partial B}, dB \rangle_F. \end{aligned} \tag{4}$$

Similarly, LoRA sets $dA = -\frac{\partial L}{\partial A} \triangleq -g_{lora}^A$ and $dB = -\frac{\partial L}{\partial B} \triangleq -g_{lora}^B$, and thus $dL = -\|\frac{\partial L}{\partial A}\|_F^2 - \|\frac{\partial L}{\partial B}\|_F^2 \le 0$. Moreover, employing the chain rule, we derive:

$$g_{lora}^A = \frac{\partial L}{\partial W} \frac{\partial W}{\partial A} = sB^T g, \qquad g_{lora}^B = \frac{\partial L}{\partial W} \frac{\partial W}{\partial B} = sgA^T. \tag{5}$$

**Why LoRA performs worse than full fine-tuning.** With Equation (3) and (4), we observe a critical connection between full fine-tuning and LoRA in the optimization process. In LoRA, changes in matrices $A$ and $B$ are inherently linked to changes in matrix $W$, i.e., $dW = \frac{\partial W}{\partial A}^T dA + \frac{\partial W}{\partial B}^T dB$. This indicates that updating $A$ and $B$ with gradient $g^A$ and $g^B$ is equivalent to performing full fine-tuning on $W$ via the following update:

$$dW = \frac{\partial W}{\partial A}^T dA + \frac{\partial W}{\partial B}^T dB = -(sBg^A + sg^B A). \tag{6}$$

Equation (6) reveals that LoRA optimization is equivalent to full fine-tuning using a low-rank gradient $\tilde{g} = sBg^A + sg^B A$ (which rank is at most to $2r$[1]) for optimization. Consequently, the performance gap between LoRA and full fine-tuning may stem from differences between $\tilde{g}$ and the full gradient $g$. The low-rank gradient $\tilde{g}$ may lose important information contained in $g$, leading to distinct optimization trajectories and ultimately causing LoRA to converge to a sub-optimal solution.

## 2.3 LOW-RANK ADAPTATION WITH EQUIVALENT GRADIENT

**Definition 1** (Equivalent Gradient). *In the context of LoRA optimization, we define the equivalent gradient as,*
$$\tilde{g} \triangleq sBg^A + sg^B A,$$
*where $s$ is the scaling factor, and $g^A$ and $g^B$ are gradients with respect to $A$ and $B$, respectively.*

In this part, we introduce our LoRA-Pro method, which bridges the performance gap by minimizing the discrepancy between the gradients. For convenience, we define the concept of equivalent gradient in Definition 1. Equivalent gradient describes the virtual low-rank gradient of the matrix $W$ in

---

[1]We provide the proof in Appendix B.1

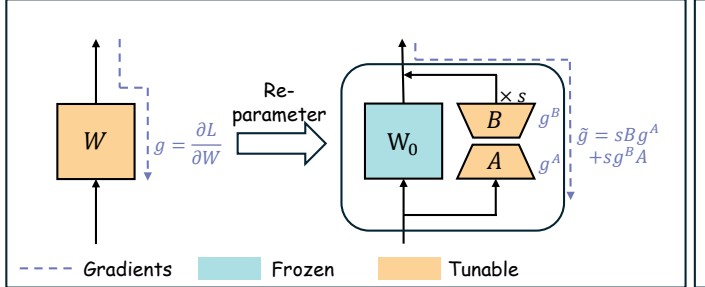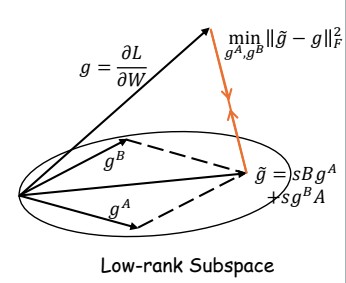

Figure 1: **Illustration of LoRA-Pro.** LoRA (Hu et al., 2022) reduces the trainable parameter by re-parameterizing the weight into the product of two low-rank matrices, i.e., $W = W_0 + sBA$. We have discovered a connection between the optimization processes of full fine-tuning and LoRA. Updating matrices $B$ and $A$ using gradients $g^B$ and $g^A$ is equivalent to updating weight $W$ using a virtual low-rank gradient $\tilde{g} = sBg^A + sg^BA$. Therefore, in LoRA-Pro, we aim to adjust gradients $g^B$ and $g^A$ to minimize the distance between the equivalent gradient $\tilde{g}$ and the full fine-tuning gradient $g$, thereby reducing their performance gap. In Theorem 2.1, we provide the optimal update gradients, and in Appendix C, we present the pseudo-code for the optimization algorithm.

LoRA optimization process, despite $W$ not being a trainable parameter. To narrow the performance gap, our goal is to carefully adjust $g^A$ and $g^B$ of matrices $A$ and $B$ to minimize the distance between the equivalent gradient $\tilde{g}$ and the full gradient $g$ in full fine-tuning. Hence, our objective is:

$$
\begin{aligned}
&\min_{g^A, g^B} \|\tilde{g} - g\|_F^2 \\
\text{s.t.} \quad &\tilde{g} = sBg^A + sg^BA, \\
&\mathrm{d}L \le 0.
\end{aligned}
\tag{7}
$$

Here, $\|\cdot\|_F$ denotes the Frobenius norm, and $\mathrm{d}L$ denotes the change in loss when updating with gradients $g^A$ and $g^B$. The objective aims to minimize the distance of the gradients while ensuring a decrease in loss using the solutions for $g^A$ and $g^B$.

**Closed-form solution.** Fortunately, we prove that the optimization problem (7) admits an optimal closed-form solution, as stated in Theorem 2.1. Additionally, an interesting observation arises from Theorem 2.1: while the full gradient $g$ serves as the ground truth in the objective, it does not necessarily explicit appear in the closed-form solution. Instead, the closed-form solution for the optimal gradients can be expressed in terms of the gradients of LoRA. This allows for an efficient gradient adjustment process, where we backpropagate using the standard LoRA and adjust the gradients of matrices $A$ and $B$ based on the closed-form solution presented in Theorem 2.1. [2]

**Theorem 2.1.** *Assume matrices $B \in \mathbb{R}^{m \times r}, A \in \mathbb{R}^{r \times n}$ are both full rank. For the objective $\min_{g^A, g^B} \|\tilde{g} - g\|_F^2$, the optimal solutions are given by:*

$$
g^A = \frac{1}{s}(B^T B)^{-1} B^T g + XA = \frac{1}{s^2}(B^T B)^{-1} g_{lora}^A + XA,
\tag{8}
$$

$$
g^B = \frac{1}{s}[I - B(B^T B)^{-1} B^T] g A^T (AA^T)^{-1} - BX
\tag{9}
$$

$$
= \frac{1}{s^2}[I - B(B^T B)^{-1} B^T] g_{lora}^B (AA^T)^{-1} - BX.
\tag{10}
$$

*Here, $X \in \mathbb{R}^{r \times r}$ represents an arbitrary matrix.*

*Proof.* See Appendix B.2. □

**Loss minimization.** While Theorem 2.1 offers a closed-form solution to the optimization problem $\min_{g^A, g^B} \|\tilde{g} - g\|_F^2$, it's crucial to understand that this solution does not inherently guarantee a

---

[2]We provide detailed algorithms in Appendix C.

decrease in loss when updating the matrices $A$ and $B$. To address this concern, we introduce Theorem 2.2. In this theorem, we prove that the change in loss $\mathrm{d}L$ can be expressed as a negative sum of two Frobenius norms, which leads to $\mathrm{d}L < 0$. This result ensures that the optimization process consistently drives towards a lower loss.

**Selection of matrix X.** Although the equivalent gradient itself is not directly related to the matrix $X$, the presence of $X$ plays a significant role in the updates of matrices $A$ and $B$. We select an appropriate $X$ such that $g^A$ and $g^B$ remain close to $g_{lora}^A$ and $g_{lora}^B$ respectively. To achieve this, we minimize their Frobenius norm, as demonstrated in Equation (14). In practical terms, $B^T B$ and $-AA^T$ do not share common eigenvalues. Therefore, according to Theorem 2.3, we can determine a unique optimal $X$ for updating matrices $A$ and $B$.

---

**Theorem 2.2.** *When updating matrices $A$ and $B$ using the closed-form solution from Theorem 2.1, we proceed as follows:*

$$A \leftarrow A - \gamma g^A \tag{11}$$

$$B \leftarrow B - \gamma g^B, \tag{12}$$

*where $\gamma \geq 0$ denotes the learning rate. Our method ensures a decrease in the loss, akin to the standard gradient descent algorithm, expressed by:*

$$\mathrm{d}L = -\gamma\{\langle g_{lora}^A, \frac{1}{s^2}(B^T B)^{-1} g_{lora}^A\rangle_F + \langle g_{lora}^B, \frac{1}{s^2}[I - B(B^T B)^{-1} B^T] g_{lora}^B (AA^T)^{-1}\rangle_F\} \leq 0. \tag{13}$$

*Proof.* See Appendix B.3. □

---

**Theorem 2.3.** *Consider the optimization problem,*

$$\min_X \|g^A - g_{lora}^A\|_F^2 + \|g^B - g_{lora}^B\|_F^2, \tag{14}$$

*where $g^A$ and $g^B$ are the optimal solutions as stated in Theorem 2.1. The optimal $X$ can be determined by solving the Sylvester equation:*

$$B^T B X + X AA^T = -\frac{1}{s^2}(B^T B)^{-1} g_{lora}^A A^T, \tag{15}$$

*which has a unique solution $X$ provided that $B^T B$ and $-AA^T$ do not have any shared eigenvalues.*

*Proof.* See Appendix B.4. □

---

## 3 EXPERIMENTAL RESULTS

In this section, we present extensive experiments to evaluate the effectiveness of LoRA-Pro across various tasks and models. First, we assess natural language understanding capabilities using the GLUE benchmark by fine-tuning the T5-base (Raffel et al., 2020) model in Section 3.1. Next, we evaluate its capabilities in dialogue generation, mathematical reasoning, and code generation using the Llama-2-7B model (Touvron et al., 2023), covered in Section 3.2. We then examine LoRA-Pro's effectiveness on image classification tasks using the CLIP-ViT-B/16 (Radford et al., 2021) model in Section 3.3. Finally, we conduct an ablation study of LoRA-Pro in Section 3.4.

**Training details.** To ensure a fair comparison, we align our experimental setup with that of LoRA-GA (Wang et al., 2024a). By default, we fine-tune the model using the AdamW optimizer (Loshchilov & Hutter, 2019) with hyper-parameters $\beta_1 = 0.9$, $\beta_2 = 0.999$, and weight decay set to 0. We implement a cosine learning rate schedule with a warmup ratio of 0.03. LoRA is applied to all linear modules, excluding the embedding layer, normalization layer, and classification head. By default, we set the rank $r = 8$ and $\alpha = 16$.

For natural language understanding tasks, we fine-tune T5-base (Raffel et al., 2020) model with learning rate 1e-4. The sequence length is set to 128, and the training batch size is 32. For dialogue generation, mathematical reasoning and code generation tasks, we fine-tune the Llama-2-7B (Touvron et al., 2023) model with a learning rate of 2e-5. We set the sequence length to 1024 and the macro batch size to 32. For image classification tasks, we fine-tune the CLIP-ViT-B/16 (Radford et al., 2021) model with learning rate 1e-4. The classifier is obtained using prompts such as "a photo of a {class}" and kept frozen during fine-tuning. And the training batch size is set to 64.

All experiments are conducted on NVIDIA RTX A6000 GPUs. To obtain a reliable estimate of model performance, we perform three runs with different random seeds and report the average and standard deviation of the results.

## 3.1 RESULTS ON NATURAL LANGUAGE UNDERSTANDING TASKS

Table 1: Results of fine-tuning T5-base using full fine-tuning and various LoRA variants on a subset of the GLUE datasets. The LoRA rank is set to 8 by default. **Bold** and underline indicate the highest and second-highest scores, respectively.

| Method | MNLI | SST2 | CoLA | QNLI | MRPC | Average |
|---|---|---|---|---|---|---|
| Full FT | **86.33±0.00** | **94.75±0.21** | 80.70±0.24 | 93.19±0.22 | 84.56±0.73 | 87.91 |
| LoRA | 85.30±0.04 | 94.04±0.11 | 69.35±0.05 | 92.96±0.09 | 68.38±0.01 | 82.08 |
| PiSSA | 85.75±0.07 | 94.07±0.06 | 74.27±0.39 | 93.15±0.14 | 76.31±0.51 | 84.71 |
| rsLoRA | 85.73±0.10 | 94.19±0.23 | 72.32±1.12 | 93.12±0.09 | 52.86±2.27 | 79.64 |
| LoRA+ | 85.81±0.09 | 93.85±0.24 | 77.53±0.20 | 93.14±0.03 | 74.43±1.39 | 84.95 |
| LoRA-GA | 85.70±0.09 | 94.11±0.18 | 80.57±0.20 | 93.18±0.06 | 85.29±0.24 | 87.77 |
| DoRA | 85.67±0.09 | 94.04±0.53 | 72.04±0.94 | 93.04±0.06 | 68.08±0.51 | 82.57 |
| AdaLoRA | 85.45±0.11 | 93.69±0.20 | 69.16±0.24 | 91.66±0.05 | 68.14±0.28 | 81.62 |
| LoRA-Pro | 86.03±0.19 | 94.19±0.13 | **81.94±0.24** | **93.42±0.05** | **86.60±0.14** | **88.44** |

In this section, we evaluate our LoRA-Pro across various natural language understanding datasets. To provide a comprehensive comparison, we include several baseline methods: 1) full fine-tuning and the standard LoRA (Hu et al., 2022). 2) LoRA variants maintaining the original structure, such as rsLoRA (Kalajdzievski, 2023), LoRA+ (Hayou et al., 2024), PiSSA (Meng et al., 2024), and LoRA-GA (Wang et al., 2024a), 3) LoRA variants with modified structures, including DoRA (Liu et al., 2024) and AdaLoRA (Zhang et al., 2023).

The results are presented in Table 1. We fine-tune the T5-base model (Raffel et al., 2020) with the baseline methods on a subset of GLUE datasets: MNLI, SST2, CoLA, QNLI, and MRPC. As shown in Table 1, we observe that LoRA-Pro achieves the highest scores on 3 out of 5 datasets and the highest average score across all 5 datasets, and achieves the highest accuracy on average over the 5 datasets. Specifically, on average over 5 datasets, LoRA-Pro surpasses standard LoRA (Hu et al., 2022) with a margin of 6.36. And LoRA-Pro even achieve higher than full fine-tuning. This superior performance may be attributed to overfitting in full fine-tuning, where optimizing all model parameters can lead to overfitting on the training data, thus reducing the model's generalization to the test set. This effect is particularly pronounced on small datasets, such as MRPC, which contains only 3.7k training data. These results validate the effectiveness of our methods.

## 3.2 RESULTS ON LARGE LANGUAGE MODELS

In this section, we evaluate the performance of LoRA-Pro on large language models, focusing on dialogue generation, mathematical reasoning, and code generation capabilities. Our experimental setup follows the configuration used in LoRA-GA (Wang et al., 2024a).

- For the dialogue generation task, we fine-tune the Llama-2-7B (Touvron et al., 2023) model on a 52k subset of the WizardLM dataset (Xu et al., 2024) and evaluate it using the MT-Bench dataset (Zheng et al., 2024a). GPT-4 is used to assess the quality of the model's responses, and we report the first-turn score as the metric.

Table 2: Fine-tuning results of Llama-2-7B model. We fine-tune the Llama-2-7B model using full fine-tuning and LoRA variants on subsets of the WizardLM (Xu et al., 2024), MetaMathQA (Yu et al., 2024), and CodeFeedback (Zheng et al., 2024b) datasets, respectively. And we assess dialogue generation, mathematical reasoning, and coding abilities on MT-Bench, GSM8K, and HumanEval datasets. **Bold** and underline indicate the highest and second-highest scores, respectively.

| Method | MT-Bench | GSM8K | HumanEval |
|---|---|---|---|
| Full FT | 5.30±0.11 | **59.36±0.85** | **35.31±2.13** |
| LoRA | 5.61±0.10 | 42.08±0.04 | 14.76±0.17 |
| PiSSA | 5.30±0.02 | 44.54±0.27 | 16.02±0.78 |
| rsLoRA | 5.25±0.03 | 45.62±0.10 | 16.01±0.79 |
| LoRA+ | 5.71±0.08 | 52.11±0.62 | 18.17±0.52 |
| DoRA | **5.97±0.02** | 53.07±0.75 | 19.75±0.41 |
| AdaLoRA | 5.57±0.05 | 50.72±1.39 | 17.80±0.44 |
| LoRA-GA | 5.95±0.16 | 53.60±0.30 | 19.81±1.46 |
| LoRA-GA (rank=32) | 5.79±0.09 | 55.12±0.30 | 20.18±0.19 |
| LoRA-GA (rank=128) | 6.13±0.07 | 55.07±0.18 | 23.05±0.37 |
| LoRA-Pro | 5.72±0.03 | 57.57±0.50 | 22.97±0.35 |
| LoRA-Pro (rank=32) | 5.57±0.51 | 57.97±0.50 | 26.63±0.35 |
| LoRA-Pro (rank=128) | 5.67±0.11 | 61.08±0.19 | 30.28±0.93 |

- For the math task, we fine-tune the Llama-2-7B (Touvron et al., 2023) model on a 100k sample from the MetaMathQA dataset (Yu et al., 2024). The model is then evaluated on the GSM8K test set (Cobbe et al., 2021), and we report the accuracy as the metric.

- For the coding task, we fine-tune the Llama-2-7B (Touvron et al., 2023) model on a 100k subset of the CodeFeedback dataset (Zheng et al., 2024b) and test it on the HumanEval dataset (Chen et al., 2021), reporting the PASS@1 metric.

We compare LoRA-Pro with several baselines, including full fine-tuning, LoRA (Hu et al., 2022), PiSSA (Meng et al., 2024), rsLoRA (Kalajdzievski, 2023), LoRA+ (Hayou et al., 2024), DoRA (Liu et al., 2024), AdaLoRA (Zhang et al., 2023), and LoRA-GA (Wang et al., 2024a). By default, we set the rank to 8 and $\alpha = 16$. Following LoRA-GA (Wang et al., 2024a), we initialize the scaling factor as in rsLoRA (Kalajdzievski, 2023), i.e., $s = \frac{\alpha}{\sqrt{r}}$.

Table 2 presents our experimental results, which demonstrate LoRA-Pro's superior performance. With a rank of 8, LoRA-Pro achieves notable improvements over the original LoRA: 0.11 on MT-Bench, 15.49 on GSM8K, and 8.21 on HumanEval. When compared to the second-best PEFT method, LoRA-GA, LoRA-Pro shows consistent gains: 3.97 on GSM8K and a substantial 3.16 on HumanEval. These results validate the effectiveness of our LoRA-Pro method.

Interestingly, we observe that full fine-tuning unexpectedly underperforms on MT-Bench. We attribute this to potential discrepancies between the WizardLM training data distribution and the MT-Bench evaluation set. The extensive learning capacity of full fine-tuning may lead to overfitting on the training distribution, compromising generalization to MT-Bench. Since LoRA-Pro aligns more closely with full fine-tuning during optimization, its relatively poor performance on MT-Bench may also be attributed to overfitting.

To further explore the scalability of our method, we increase the rank in LoRA-Pro from 8 to 128. Our observations reveal a clear trend: as the rank increases, the performance gap between LoRA-Pro and full fine-tuning narrows rapidly. Notably, LoRA-Pro consistently outperforms LoRA-GA at the same ranks on both GSM8K and HumanEval datasets. At rank 32, LoRA-Pro surpasses LoRA-GA by 2.85 on GSM8K and 6.45 on HumanEval. This performance disparity becomes even more pronounced at rank 128, where LoRA-Pro outperforms LoRA-GA by 6.01 on GSM8K and 7.23 on HumanEval. These results demonstrate the superior scalability and effectiveness of LoRA-Pro across various rank settings.

### 3.3 RESULTS ON IMAGE CLASSIFICATION TASKS

Table 3: Fine-tuning results of CLIP-ViT-B/16 on image classification tasks. We fine-tune CLIP-ViT-B/16 using full fine-tuning and LoRA variants across StanfordCars, DTD, EuroSAT, GTSRB, RESISC45, SUN397, and SVHN datasets. **Bold** indicates the highest results, while underline represents the second-highest results.

| Method | Cars | DTD | EuroSAT | GTSRB | RESISC45 | SUN397 | SVHN | Average |
|---|---|---|---|---|---|---|---|---|
| Zero-shot | 63.75 | 44.39 | 42.22 | 35.22 | 56.46 | 62.56 | 15.53 | 45.73 |
| Full FT | 84.23±0.06 | 77.44±0.19 | 98.09±0.03 | 94.31±0.28 | 93.95±0.0 | 75.35±0.10 | 93.04±0.18 | 88.06 |
| LoRA | 72.81±0.13 | 73.92±0.38 | 96.93±0.07 | 92.40±0.10 | 90.03±0.14 | 70.12±0.18 | 88.02±0.07 | 83.46 |
| rsLoRA | 82.38±0.20 | 78.03±0.76 | 98.06±0.08 | 95.04±0.11 | 93.96±0.18 | 75.38±0.24 | 92.74±0.18 | 87.94 |
| LoRA+ | 72.87±0.18 | 74.07±0.45 | 97.01±0.02 | 92.42±0.18 | 89.96±0.11 | 70.17±0.15 | 88.08±0.05 | 83.51 |
| DoRA | 73.72±0.06 | 73.72±0.33 | 96.95±0.01 | 92.38±0.17 | 90.03±0.08 | 70.20±0.19 | 88.23±0.05 | 83.48 |
| LoRA-GA | 85.18±0.41 | 77.50±0.12 | 98.05±0.27 | 95.28±0.10 | 94.43±0.19 | 75.44±0.06 | 93.68±0.35 | 88.51 |
| LoRA-Pro | **85.87±0.08** | **78.64±0.25** | **98.46±0.03** | **95.66±0.05** | **94.75±0.21** | **76.42±0.14** | **94.63±0.20** | **89.20** |

In this section, we assess the performance of LoRA-Pro on image classification tasks. To provide a comprehensive comparison, we compare it with several baselines: zero-shot CLIP (Radford et al., 2021), full fine-tuning, vanilla LoRA (Hu et al., 2022), rsLoRA (Kalajdzievski, 2023), LoRA+ (Hayou et al., 2024), DoRA (Liu et al., 2024), and LoRA-GA (Wang et al., 2024a).

We fine-tune the CLIP-ViT-B/16 (Radford et al., 2021) model on various datasets, including StanfordCars (Krause et al., 2013), DTD (Cimpoi et al., 2014), EuroSAT (Helber et al., 2019), GT-SRB (Houben et al., 2013), RESISC45 (Cheng et al., 2017), SUN397 (Xiao et al., 2010), and SVHN (Netzer et al., 2011). Accuracy is used as the evaluation metric. During fine-tuning, only the visual backbone of the CLIP-ViT-B/16 model is updated, while the classifier, derived from prompts such as "a photo of a {class}", remains frozen.

The results are presented on Table 3. LoRA-Pro achieves the highest accuracy across all seven datasets. Specifically, on average, LoRA-Pro surpasses zero-shot classification by 43.47, outperforms LoRA (Hu et al., 2022) by 5.74, and exceeds rsLoRA (Kalajdzievski, 2023) by 1.26. These results validate the effectiveness of our LoRA-Pro method.

### 3.4 ABLATION STUDY

**Ablation study on the full-rank assumption.** In Theorem 2.1, we assume that the matrices $A \in \mathbb{R}^{r \times n}$ and $B \in \mathbb{R}^{m \times r}$ are full-rank during training. Our goal here is to verify whether this assumption holds in practice. We track the rank changes of all $A$ and $B$ matrices during the fine-tuning process of Llama-2-7B on the Meta-MathQA (Yu et al., 2024) dataset.

In Figure 2, we illustrate the rank changes of matrices $A$ and $B$ from the $q$ projection of layer 9 during the training process, with rank set to 8 and 32, respectively. We observed that, although $A$ and $B$ do not initially satisfy the full rank assumption (with matrix $B$ initialized as a zero matrix), both matrices achieve full rank after the first update step. The rank behavior of $A$ and $B$ in other layers also exhibits similar results.

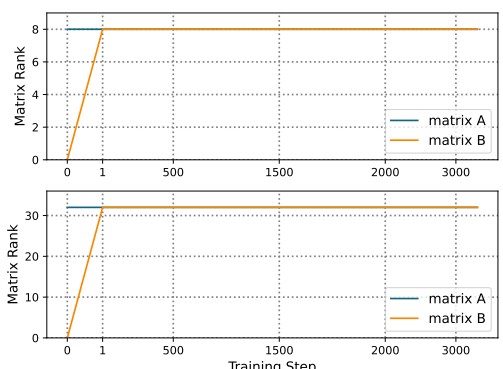

Figure 2: Visualization of matrix ranks of A and B during training, with ranks set to 8 and 32, respectively.

This observation provides practical evidence that the assumption in Theorem 2.1 is reasonable and supports the validity of the proposed solutions.

**Memory footprint and training time.** Here, we evaluate the additional costs associated with using LoRA-Pro compared to LoRA. We primarily focus on comparing the differences in memory cost and training time between LoRA-Pro, LoRA, and full fine-tuning. The results of the experiments are presented in Table 4. We measure memory cost using 8 A6000 GPUs with a batch size of 1.

Table 4: We compare LoRA, LoRA-Pro, and Full Fine-Tuning in terms of memory cost, training time, and performance on the MT-Bench, GSM8K, and HumanEval datasets. Memory cost is measured using a single A6000 GPU with a batch size of 1. Training time is recorded on the Meta-MathQA dataset using 8 A100 GPUs with DeepSpeed ZeRO-2 stage optimization.

| Method | Memory Cost | Training Time | MT-Bench | GSM8K | HumanEval |
|--------|-------------|---------------|----------|-------|-----------|
| Full FT | $\sim 8 \times 40$ GB | 4h 20min | 5.30±0.11 | 59.36±0.85 | 35.31±2.13 |
| LoRA | $\sim 8 \times 17$ GB | 1h 30min | 5.61±0.10 | 42.08±0.04 | 14.76±0.17 |
| LoRA-GA | $\sim 8 \times 17$ GB | 1h 31min | 5.95±0.16 | 53.60±0.30 | 19.81±1.46 |
| LoRA-Pro | $\sim 8 \times 21$ GB | 1h 41min | 5.72±0.03 | 57.57±0.50 | 22.97±0.35 |

Training time is recorded on the WizardLM dataset using 8 A100 GPUs with DeepSpeed (Rasley et al., 2020) ZeRO-2 stage optimization.

The results are shown in Table 4. From the table, we observe the following: 1) LoRA-Pro requires approximately 4GB more GPU memory compared to LoRA. This difference likely stems from the need to compute $B^T B$, $AA^T$, and their inverses during the calculation of the optimal gradients. 2) Surprisingly, the training time for LoRA-Pro is nearly identical to that of LoRA, with only about 10 minutes increase in additional training time. We attribute this to the fact that the matrices $A$ and $B$ in LoRA are low-rank. Consequently, the extra computations required by LoRA-Pro (such as matrix inversion and the calculation of $X$) are performed on small $r \times r$ matrices, making the extra computational overhead manageable.

Considering that LoRA-Pro uses less memory and trains faster than full fine-tuning, while also providing performance improvements over LoRA, we believe that the additional memory and training time costs are acceptable.

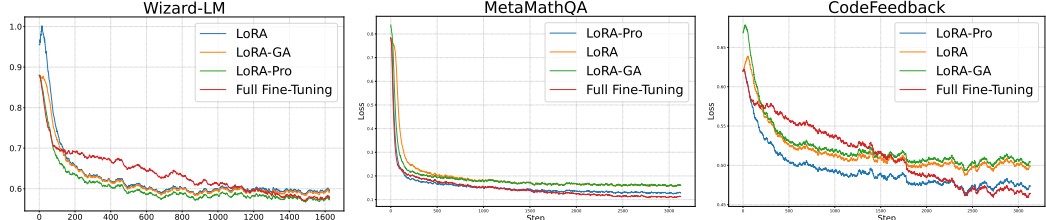

Figure 3: Training loss curves of LoRA, LoRA-GA, LoRA-Pro, and Full Fine-tuning on WizardLM, MetaMathQA, and CodeFeedback.

**Training curves of LoRA-Pro.** In this part, we present the training loss curves for LoRA, LoRA-GA, LoRA-Pro, and Full Fine-tuning across WizardLM, MetaMathQA, and CodeFeedback datasets. As illustrated in Figure 3, LoRA-Pro demonstrates a faster convergence speed compared to LoRA and LoRA-GA. Furthermore, LoRA-Pro achieves a lower final loss value upon convergence, indicating its improved efficiency and effectiveness.

## 4 RELATED WORK

**Parameter-Efficient Fine-Tuning.** Given the huge size of foundation models, recent research has focused on developing parameter-efficient fine-tuning methods (Hu et al., 2022; Liu et al., 2024; Ding et al., 2023; Houlsby et al., 2019; Liu et al., 2023; Lester et al., 2021; Wang et al., 2024c). These methods aim to reduce the cost of fine-tuning by adjusting only a small portion of the model's parameters. Generally, these methods fall into two main categories. The first category is adapter tuning (Houlsby et al., 2019; Sung et al., 2022; He et al., 2021; Zhang et al., 2024; Bapna & Firat, 2019; Hu et al., 2022), which involves inserting small neural network modules, called adapters, into specific layers of the model. During fine-tuning, we keep the model frozen and only fine-tune the lightweight adapter modules, significantly reducing the memory footprint for fine-tuning. The second category is prompt tuning (Lester et al., 2021; Zhou et al., 2022; Li & Liang, 2021; Liu et al., 2022; Wang et al., 2023; 2024b; Liang et al., 2024). Prompt tuning adapts the models to specific

tasks by adding specially designed prompts or learnable tokens to the input data, rather than directly modifying the internal parameters of foundation models. In this paper, we focus on LoRA (Hu et al., 2022), a prominent method within the realm of adapter tuning.

**Low-Rank Adaptation.** Low-rank adaptation, initially referred to as LoRA (Hu et al., 2022), has evolved into a broad category encompassing parameter-efficient fine-tuning methods based on low-rank approximations (Hu et al., 2022; Liu et al., 2024; Hayou et al., 2024; Kalajdzievski, 2023; Zhang et al., 2023; Kopiczko et al., 2024; Hyeon-Woo et al., 2022; Zhang & Pilanci, 2024; Wang et al., 2024a; Zhao et al., 2024; Wang et al., 2024a). LoRA (Hu et al., 2022) assumes that the changes in the weights of pre-trained models exhibit a low-rank structure. Consequently, it re-parameterizes these changes as the product of low-rank matrices, thereby reducing the cost during fine-tuning.

Several variants of LoRA have been proposed to address different aspects of this approach. For example, DoRA (Liu et al., 2024) improves LoRA (Hu et al., 2022) by incorporating a learnable magnitude vector to re-scale the normalized product of low-rank matrices. Another variant, rsLoRA (Kalajdzievski, 2023), introduces a new scaling factor to stabilize training in high-rank scenarios. LoRA+ (Hayou et al., 2024) improves upon LoRA by applying different learning rates to the two low-rank matrices. Additionally, Galore (Zhao et al., 2024) employs SVD to project the gradients and its first and second momentum of full training into a low-rank space, thereby reducing the memory footprint during pre-training and fine-tuning.

## 5 CONCLUSION

In this paper, we introduce LoRA-Pro, a novel approach designed to bridge the performance gap between LoRA and full fine-tuning. We have discovered that using LoRA for fine-tuning is equivalent to fine-tuning the original weights with a virtual equivalent low-rank gradient. Based on this insight, we propose adjusting the gradients of matrices $A$ and $B$ to make the equivalent gradient match the true full fine-tuning gradient, thereby reducing their performance gap. Fortunately, we theoretically prove that there exists an optimal closed-form solution for updating matrices $A$ and $B$, which are applied during fine-tuning in LoRA-Pro. To validate the effectiveness of our method, we conduct extensive experiments across various domains, including natural language understanding, dialogue generation, mathematical reasoning, code generation, and image classification tasks. The results demonstrate that LoRA-Pro significantly improves LoRA performance and narrows the performance gap with full fine-tuning.

**Limitations**. LoRA-Pro still have some limitations: (1) LoRA-Pro still adheres to LoRA's assumption that $\Delta W$ is of low rank. However, this assumption may break down in cases of pre-training or when there is a large amount of fine-tuning data, potentially leading to suboptimal results. (2) So far, we have only applied LoRA-Pro to variants that have a structure similar to LoRA. It currently cannot be applied to structurally different LoRA models, such as DoRA (Liu et al., 2024) or FLoRA (Wen & Chaudhuri, 2024). We plan to explore these directions in future research.

## ACKNOWLEDGEMENT

This work was funded by the National Natural Science Foundation of China under Grants (62276256, U2441251) and the Young Elite Scientists Sponsorship Program by CAST (2023QNRC001). We thank Jie Cheng and Yongcan Yu for providing computational resources critical to this work.

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

# LoRA-Pro: Are Low-Rank Adapters Properly Optimized?

## ————Appendix————

The structure of Appendix is as follows,

- Appendix A contains the notation usage in our paper.
- Appendix B contains the proofs of the theorems in the main manuscript.
- Appendix C details the optimization algorithms of the proposed method.
- Appendix D presents additional experimental results.

## A   NOTATION

In Table 5, we detail the notations utilized in our paper.

Table 5: Description of notations used in the paper.

| Notation | Description |
|---|---|
| $s$ | scaling factor in LoRA |
| $B \in \mathbb{R}^{m \times r}, A \in \mathbb{R}^{r \times n}$ | low rank matrices in LoRA |
| $g = \frac{\partial L}{\partial W} \in \mathbb{R}^{m \times n}$ | gradients of full fine-tuning |
| $g_{lora}^A = \frac{\partial L}{\partial A} = sB^T g \in \mathbb{R}^{r \times n}$ | gradients of matrix A in LoRA |
| $g_{lora}^B = \frac{\partial L}{\partial B} = sgA^T \in \mathbb{R}^{m \times r}$ | gradients of matrix B in LoRA |
| $\mathrm{d}L$ | differential of the loss function |
| $\mathrm{d}A$ | differential of the matrix A |
| $\mathrm{d}B$ | differential of the matrix B |
| $\|\cdot\|_F$ | Frobenius Norm |
| $\langle\cdot,\cdot\rangle_F$ | Frobenius inner product |

## B   PROOF OF THEORETICAL RESULTS

### B.1   PROOF THAT THE EQUIVALENT GRADIENT IS LOW-RANK

**Lemma.** *Assume $B \in \mathbb{R}^{m \times r}, A \in \mathbb{R}^{r \times n}$ and $g^B \in \mathbb{R}^{m \times r}, g^A \in \mathbb{R}^{r \times n}$ represent matrices and their corresponding gradients in LoRA optimization. We demonstrate that the equivalent gradient:*

$$\tilde{g} = sg^B A + sBg^A, \tag{16}$$

*where $s > 0$ is the scaling factor, has matrix rank at most $2r$.*

*Proof.* Since matrix rank satisfies the property of subadditivity, we have:

$$rank(\tilde{g}) = rank(sg^B A + sBg^A) \leq rank(g^B A) + rank(Bg^A). \tag{17}$$

Furthermore, for any matrices $A$ and $B$, $rank(AB) \leq \min(rank(A), rank(B))$. Therefore, we can bound the ranks as follows:

$$rank(g^B A) \leq \min(rank(g^B), rank(A)) \leq r \tag{18}$$

$$rank(Bg^A) \leq \min(rank(B), rank(g^A)) \leq r \tag{19}$$

Thus, in conclusion, the equivalent gradient has a rank of at most $2r$:

$$rank(\tilde{g}) \leq rank(g^B A) + rank(Bg^A) \tag{20}$$

$$\leq \min(rank(g^B), rank(A)) + \min(rank(B), rank(g^A)) \tag{21}$$

$$\leq 2r. \tag{22}$$

$\square$

## B.2 PROOF OF THEOREM 2.1

**Theorem.** *Assume matrices $B \in \mathbb{R}^{m \times r}, A \in \mathbb{R}^{r \times n}$ are both full rank. For the objective $\min_{g^A, g^B} \|\tilde{g} - g\|_F^2$, the solutions are given by:*

$$g^A = \frac{1}{s}(B^T B)^{-1} B^T g + XA = \frac{1}{s^2}(B^T B)^{-1} g_{lora}^A + XA \tag{23}$$

$$g^B = \frac{1}{s}[I - B(B^T B)^{-1} B^T] g A^T (AA^T)^{-1} - BX \tag{24}$$

$$= \frac{1}{s^2}[I - B(B^T B)^{-1} B^T] g_{lora}^B (AA^T)^{-1} - BX. \tag{25}$$

*Here, $X \in \mathbb{R}^{r \times r}$ represents an arbitrary matrix.*

*Proof.* For simplicity, we denote $L = \|sBg^A + sg^B A - g\|_F^2$. To solve the optimization problem, we need to satisfy the following conditions:

$$\frac{\partial L}{\partial g^A} = 2sB^T(sBg^A + sg^B A - g) = 0 \tag{26}$$

$$\frac{\partial L}{\partial g^B} = 2(sBg^A + sg^B A - g)sA^T = 0 \tag{27}$$

Given that matrices $A$ and $B$ are full-rank, $AA^T$ and $B^T B$ are invertible. And from Equation (27), we derive:

$$g^B = \frac{1}{s} g A^T (AA^T)^{-1} - Bg^A A^T (AA^T)^{-1}. \tag{28}$$

Substituting this into Equation (26), we obtain the following linear equation:

$$g^A[I - A^T(AA^T)^{-1}A] = \frac{1}{s}(B^T B)^{-1} B^T g[I - A^T(AA^T)^{-1}A]. \tag{29}$$

Here, we notice that the matrix $P = I - A^T(AA^T)^{-1}A$ is a projection matrix with rank $n - r$. The solution to the linear equation (29) is:

$$g^A = \frac{1}{s}(B^T B)^{-1} B^T g + XA, \tag{30}$$

where $X \in \mathbb{R}^{r \times r}$ represents an arbitrary matrix. We take the solution (30) into Equation (28), we derive:

$$g^B = \frac{1}{s}[I - B(B^T B)^{-1} B^T] g A^T (AA^T)^{-1} - BX \tag{31}$$

While we have obtained closed-form solutions for $g^A$ and $g^B$, these solutions explicitly depend on the gradient of the matrix $W$, i.e., $g$, which is undesirable since $g$ is unknown during LoRA optimization. Fortunately, the solutions can be transformed into the forms of the gradients of standard LoRA, where the gradients are:

$$g_{lora}^A = sB^T g, \quad g_{lora}^B = sg A^T. \tag{32}$$

Therefore, the solutions to the optimization problem can be written as:

$$g^A = \frac{1}{s^2}(B^T B)^{-1} g_{lora}^A + XA, \tag{33}$$

$$g^B = \frac{1}{s^2}[I - B(B^T B)^{-1} B^T] g_{lora}^B (AA^T)^{-1} - BX. \tag{34}$$

In our method, we perform the standard forward and backward passes of LoRA, then adjust the gradients of A and B using Solutions (33) and (34), and subsequently update them. $\square$

### B.3 PROOF OF THEOREM 2.2

**Theorem.** *When updating matrices $A$ and $B$ using the closed-form solution from Theorem 2.1, we proceed as follows:*

$$A \leftarrow A - \gamma g^A, \tag{35}$$

$$B \leftarrow B - \gamma g^B, \tag{36}$$

*where $\gamma \geq 0$ denotes the learning rate. Our method ensures a decrease in the loss, akin to the standard gradient descent algorithm, expressed by:*

$$\mathrm{d}L = -\gamma\{\langle g_{lora}^A, \frac{1}{s^2}(B^TB)^{-1}g_{lora}^A\rangle_F + \langle g_{lora}^B, \frac{1}{s^2}[I - B(B^TB)^{-1}B^T]g_{lora}^B(AA^T)^{-1}\rangle_F\} \leq 0 \tag{37}$$

*Proof.* In summary, the proof of Theorem 2.2 is divided into two distinct parts. To begin with, we demonstrate that $\mathrm{d}L$ can be expressed in the following form:

$$\mathrm{d}L = -\gamma\{\langle g_{lora}^A, \frac{1}{s^2}(B^TB)^{-1}g_{lora}^A\rangle_F + \langle g_{lora}^B, \frac{1}{s^2}[I - B(B^TB)^{-1}B^T]g_{lora}^B(AA^T)^{-1}\rangle_F\}. \tag{38}$$

In the second part, we prove that this expression for $\mathrm{d}L$ is always less than or equal to zero: $\mathrm{d}L \leq 0$.

**Part I.** Therefore, in this part, we first prove Equation (38). During the optimization process, the differential change in the loss function, $\mathrm{d}L$, can be expressed in terms of the differentials $\mathrm{d}A$ and $\mathrm{d}B$ as follows:

$$\mathrm{d}L = \langle \frac{\partial L}{\partial A}, \mathrm{d}A\rangle_F + \langle \frac{\partial L}{\partial B}, \mathrm{d}B\rangle_F. \tag{39}$$

From Equation (35) and (36), we can derive that:

$$\mathrm{d}A = -\gamma g^A, \quad \mathrm{d}B = -\gamma g^B. \tag{40}$$

Given that $\frac{\partial L}{\partial A} = g_{lora}^A$ and $\frac{\partial L}{\partial B} = g_{lora}^B$, it follows that:

$$\begin{aligned}
\mathrm{d}L &= -\gamma(\langle g_{lora}^A, g^A\rangle_F + \langle g_{lora}^B, g^B\rangle_F) \\
&= -\gamma(\langle g_{lora}^A, \frac{1}{s^2}(B^TB)^{-1}g_{lora}^A\rangle_F + \langle g_{lora}^B, \frac{1}{s^2}[I - B(B^TB)^{-1}B^T]g_{lora}^B(AA^T)^{-1}\rangle_F \\
&\quad + \langle g_{lora}^A, XA\rangle_F - \langle g_{lora}^B, BX\rangle_F).
\end{aligned} \tag{41}$$

And we have the following equation:

$$\begin{aligned}
&\langle g_{lora}^A, XA\rangle_F - \langle g_{lora}^B, BX\rangle_F \\
=& \langle g_{lora}^A A^T, X\rangle_F - \langle B^T g_{lora}^B, X\rangle_F \\
=& \langle g_{lora}^A A^T - B^T g_{lora}^B, X\rangle_F \\
=& \langle (sB^Tg)A^T - B^T(sgA^T), X\rangle_F \\
=& 0.
\end{aligned} \tag{42}$$

Therefore, we have:

$$\mathrm{d}L = -\gamma\{\langle g_{lora}^A, \frac{1}{s^2}(B^TB)^{-1}g_{lora}^A\rangle_F + \langle g_{lora}^B, \frac{1}{s^2}[I - B(B^TB)^{-1}B^T]g_{lora}^B(AA^T)^{-1}\rangle_F\}. \tag{43}$$

**Part II.** In this part, we aim to prove $\mathrm{d}L \leq 0$. Given that the learning rate $\gamma > 0$, it suffices to show the following inequalities:

$$\langle g_{lora}^A, \frac{1}{s^2}(B^TB)^{-1}g_{lora}^A\rangle_F \geq 0, \tag{44}$$

$$\langle g_{lora}^B, \frac{1}{s^2}[I - B(B^TB)^{-1}B^T]g_{lora}^B(AA^T)^{-1}\rangle_F \geq 0. \tag{45}$$

By proving these inequalities, we can establish that $\mathrm{d}L \leq 0$ as derived from Equation (38).

① Proof of $\langle g_{lora}^A, \frac{1}{s^2}(B^T B)^{-1} g_{lora}^A \rangle_F \geq 0$.

To begin with, we need to show that $(B^T B)^{-1}$ is positive definite. To establish this, it is sufficient to show that $B^T B$ is positive definite, as the inverse of a positive definite matrix is also positive definite. To achieve this, consider any non-zero vector $x$, and noting that $B$ is full-rank, we have,

$$\langle x, B^T B x \rangle = \langle Bx, Bx \rangle = \|Bx\|^2 > 0. \tag{46}$$

This shows that $B^T B$ is positive definite. Consequently, $(B^T B)^{-1}$ is positive definite as well. Since $(B^T B)^{-1}$ is positive definite, and thus we can apply Cholesky decomposition, and $(B^T B)^{-1} = UU^T$. With this, we have,

$$
\begin{aligned}
\langle g_{lora}^A, \frac{1}{s^2}(B^T B)^{-1} g_{lora}^A \rangle_F &= \frac{1}{s^2} \langle g_{lora}^A, UU^T g_{lora}^A \rangle_F \\
&= \frac{1}{s^2} \langle U^T g_{lora}^A, U^T g_{lora}^A \rangle_F \\
&= \frac{1}{s^2} \|U^T g_{lora}^A\|_F^2 \geq 0
\end{aligned}
\tag{47}
$$

② Proof of $\langle g_{lora}^B, \frac{1}{s^2}[I - B(B^T B)^{-1} B^T] g_{lora}^B (AA^T)^{-1} \rangle_F \geq 0$.

Similarly, we can prove that matrix $(AA^T)^{-1}$ is positive-definite. By employing Cholesky decomposition, we express $(AA^T)^{-1} = UU^T$, where $U$ is a lower-triangle matrix. Subsequently, we define $P = I - B(B^T B)^{-1} B^T$. It can be shown that $P^2 = P$ and $P$ is symmetry, indicating that $P$ is a projection matrix. Consequently, the eigenvalues of $P$ are either 0 or 1, which implies that $P$ is positive semi-definite. Utilizing the Cholesky decomposition, we derive that $P = VV^T$, where $V$ is a lower-triangle matrix. Finally, we have:

$$
\begin{aligned}
\langle g_{lora}^B, \frac{1}{s^2}[I - B(B^T B)^{-1} B^T] g_{lora}^B (AA^T)^{-1} \rangle_F &= \frac{1}{s^2} \langle g_{lora}^B, VV^T g_{lora}^B UU^T \rangle_F \\
&= \frac{1}{s^2} \langle V^T g_{lora}^B U, V^T g_{lora}^B U \rangle_F \\
&= \frac{1}{s^2} \|V^T g_{lora}^B U\|_F^2 \geq 0
\end{aligned}
\tag{48}
$$

In summary, based on the above proofs, we have demonstrated that:

$$\mathrm{d}L = -\gamma \{ \underbrace{\langle g_{lora}^A, \frac{1}{s^2}(B^T B)^{-1} g_{lora}^A \rangle_F}_{\geq 0 \text{ as shown in } ①} + \underbrace{\langle g_{lora}^B, \frac{1}{s^2}[I - B(B^T B)^{-1} B^T] g_{lora}^B (AA^T)^{-1} \rangle_F}_{\geq 0 \text{ as shown in } ②} \} \leq 0$$

$$\tag{49}$$

$\square$

### B.4 Proof of Theorem 2.3

> **Theorem.** *Consider the optimization problem,*
>
> $$\min_X \|g^A - g_{lora}^A\|_F^2 + \|g^B - g_{lora}^B\|_F^2, \tag{50}$$
>
> *where $g^A$ and $g^B$ are the optimal solutions as stated in Theorem 2.1. The optimal $X$ can be determined by solving the Sylvester equation:*
>
> $$B^T BX + XAA^T = -\frac{1}{s^2}(B^T B)^{-1} g_{lora}^A A^T, \tag{51}$$
>
> *which has a unique solution $X$ provided that $B^T B$ and $-AA^T$ do not have any shared eigenvalues.*

*Proof.* For simplicity, we denote $L = \|g^A - g_{lora}^A\|_F^2 + \|g^B - g_{lora}^B\|_F^2$. To solve the optimization problem, we need to satisfy the following conditions:

$$\frac{\partial L}{\partial X} = 0. \tag{52}$$

Since $g^A$ and $g^B$ are solutions in Theorem 2.1 and $g^A_{lora} = sB^T g$ and $g^B_{lora} = sg A^T$, we obtain that:

$$
\begin{aligned}
2(g^A - g^A_{lora})A^T - 2B^T(g^B - g^B_{lora}) &= 0, \\
\Rightarrow \quad g^A A^T - B^T g^B &= g^A_{lora} A^T - B^T g^B_{lora}, \\
\Rightarrow \quad B^T B X + X A A^T &= -\frac{1}{s^2}(B^T B)^{-1} g^A_{lora} A^T,
\end{aligned}
\tag{53}
$$

which is a Sylvester equation. This equation has a unique solution for $X$ if and only if $B^T B$ and $-AA^T$ have no shared eigenvalues.

$\square$

## C  Optimization Algorithms

In this section, we present the pseudo-codes for implementing our LoRA-Pro method using the SGD (Sutskever et al., 2013) and AdamW (Loshchilov & Hutter, 2019) optimizers. The details are provided in Algorithm 1 and Algorithm 2, respectively.

**LoRA-Pro with SGD optimizer.** In the standard SGD algorithm, as illustrated in Algorithm 1, all we need to do is adjusting the gradients of matrices $A$ and $B$ with the solutions in Theorem 2.1.

---

**Algorithm 1** LoRA-Pro with SGD optimizer

---

**Require:** Given initial learning rate $\gamma$, scaling factor $s$.
1: Initialize time step $t \leftarrow 0$, low-rank matrices $A_0 \in \mathbb{R}^{r \times n}$ and $B_0 \in \mathbb{R}^{m \times r}$
2: **repeat**
3:     $t \leftarrow t + 1$
4:     $g^A_{lora}, g^B_{lora} \leftarrow \text{SelectBatch}(A_{t-1}, B_{t-1})$     ▷ *Select batch and return the corresponding gradients*
5:     $A, B \leftarrow A_{t-1}, B_{t-1}$     ▷ *Obtain the low-rank matrices A and B*
6:     $X \leftarrow \text{SolveSylvester}(B^T B X + X A A^T = -\frac{1}{s^2}(B^T B)^{-1} g^A_{lora} A^T)$ ▷ *Compute X by solving the sylvester equation*
7:     $g^A = \frac{1}{s^2}(B^T B)^{-1} g^A_{lora} + XA$     ▷ *Adjust the gradients of LoRA with Theorem 2.1*
8:     $g^B = \frac{1}{s^2}[I - B(B^T B)^{-1}B^T]g^B_{lora}(AA^T)^{-1} - BX$
9:     $A_t \leftarrow A_{t-1} - \gamma g^A$
10:     $B_t \leftarrow B_{t-1} - \gamma g^B$
11: **until** *stopping criterion is met*
12: **return** optimized parameters $A_t$ and $B_t$

---

**LoRA-Pro with AdamW optimizer.** In AdamW optimizer, the implementation becomes more complex. We aim to closely approximate full fine-tuning during optimization. Several modifications are necessary. Firstly, in order to mimic full fine-tuning, after adjusting the gradients of matrices $A$ and $B$, we need to compute the equivalent gradient,

$$
\tilde{g} = sg^B A + sB g^A.
\tag{54}
$$

Subsequently, we calculate the first and second moments of this equivalent gradient to derive the corresponding AdamW gradient, $\tilde{g}^{AdamW}$. Secondly, we determine the gradients with respect to matrices $A$ and $B$ as follows:

$$
\tilde{g}^A = sB^T \tilde{g}^{AdamW}, \quad \tilde{g}^B = s\tilde{g}^{AdamW} A^T.
\tag{55}
$$

Thirdly, the weight decay process must be adjusted. In line with full fine-tuning, the weight decay is given by:

$$
W \leftarrow (1 - \gamma\lambda)(W_0 + sBA).
\tag{56}
$$

This can be decomposed into:

$$
W_0 \leftarrow (1 - \gamma\lambda)W_0, \quad B \leftarrow \sqrt{1 - \gamma\lambda}B, \quad A \leftarrow \sqrt{1 - \gamma\lambda}A
\tag{57}
$$

---

**Algorithm 2** LoRA-Pro with AdamW optimizer

---

**Require:** Given initial learning rate $\gamma$, scaling factor $s$, original weight matrix $W_0 \in \mathbb{R}^{m \times n}$, and $\beta_1 = 0.9, \beta_2 = 0.999, \epsilon = 10^{-8}, \lambda \in \mathbb{R}$
1: Initialize time step $t \leftarrow 0$, low-rank matrices $A_0 \in \mathbb{R}^{r \times n}$ and $B_0 \in \mathbb{R}^{m \times r}$, first momentum $m_0 \in \mathbb{R}^{m \times n}$, second momentum $v_t \in \mathbb{R}^{m \times n}$
2: **repeat**
3: $\quad t \leftarrow t + 1$
4: $\quad g^A_{lora}, g^B_{lora} \leftarrow \text{SelectBatch}(A_{t-1}, B_{t-1})$ $\qquad \triangleright$ *Select batch and return the corresponding gradients*
5: $\quad A, B \leftarrow A_{t-1}, B_{t-1}$ $\qquad\qquad\qquad\qquad \triangleright$ *Obtain the low-rank matrices A and B*
6: $\quad X \leftarrow 0$ $\qquad\qquad\qquad\qquad\qquad \triangleright$ *X's value does not affect equivalent gradient*
7: $\quad g^A = \frac{1}{s^2}(B^T B)^{-1} g^A_{lora} + XA$ $\qquad \triangleright$ *Adjust the gradients of LoRA with Theorem 2.1*
8: $\quad g^B = \frac{1}{s^2}[I - B(B^T B)^{-1}B^T]g^B_{lora}(AA^T)^{-1} - BX$
9: $\quad \tilde{g} \leftarrow sg^B A + sBg^A$ $\qquad\qquad\qquad\qquad \triangleright$ *Compute equivalent gradient*
10: $\quad m_t \leftarrow \beta_1 m_{t-1} + (1 - \beta_1)\tilde{g}$
11: $\quad v_t \leftarrow \beta_2 v_{t-1} + (1 - \beta_2)\tilde{g}^2$
12: $\quad \hat{m}_t \leftarrow \frac{m_t}{1-\beta_1^t}$
13: $\quad \hat{v}_t \leftarrow \frac{v_t}{1-\beta_2^t}$
14: $\quad \tilde{g}^{AdamW} \leftarrow \frac{\hat{m}_t}{\sqrt{\hat{v}_t}+\epsilon}$
15: $\quad \tilde{g}^A_{lora} \leftarrow sB^T \tilde{g}^{AdamW}$
16: $\quad \tilde{g}^B_{lora} \leftarrow s\tilde{g}^{AdamW} A^T$
17: $\quad X \leftarrow \text{SolveSylvester}(B^T BX + XAA^T = -\frac{1}{s^2}(B^T B)^{-1}\tilde{g}^A_{lora}A^T)$ $\triangleright$ *Compute X by solving the sylvester equation*
18: $\quad \tilde{g}^A = \frac{1}{s^2}(B^T B)^{-1}\tilde{g}^A_{lora} + XA$ $\qquad \triangleright$ *Adjust the gradients of LoRA with Theorem 2.1*
19: $\quad \tilde{g}^B = \frac{1}{s^2}[I - B(B^T B)^{-1}B^T]\tilde{g}^B_{lora}(AA^T)^{-1} - BX$
20: $\quad A \leftarrow \sqrt{1 - \gamma\lambda}A$ $\qquad\qquad\qquad\qquad\qquad \triangleright$ *Weight Decay*
21: $\quad B \leftarrow \sqrt{1 - \gamma\lambda}B$
22: $\quad W_0 \leftarrow (1 - \gamma\lambda)W_0$
23: $\quad A_t \leftarrow A_{t-1} - \gamma\tilde{g}^A$
24: $\quad B_t \leftarrow B_{t-1} - \gamma\tilde{g}^B$
25: **until** *stopping criterion is met*
26: **return** optimized parameters $A_t$ and $B_t$

---

# D    ADDITIONAL EXPERIMENTS

## D.1    ABLATION STUDY OF THE SELECTION OF X

Based on Theorem 2.1, in LoRA-Pro, the matrix $X$ can be chosen arbitrarily. While its selection does not affect the equivalent gradient, it does influence the updates of matrices $A$ and $B$ in LoRA. Here, we conduct an ablation study on the choice of $X$.

We compare three possible values for $X$. 1) Zero solution: In this simplest case, we set $X = \mathbf{0}$. 2) Sylvester solution: Here, $X$ is obtained by solving the Sylvester equation, as described in Theorem 2.3. 3) Symmetry solution: This approach aims to balance the contributions of both terms in the equation $\tilde{g} = sg^B A + sBg^A$, enforcing the condition $g^B A = Bg^A$. For the symmetry solution, solving for $X$ yields:

$$X = -\frac{1}{2s}(B^T B)^{-1} B^T g A (A^T A)^{-1} = -\frac{1}{2s^2}(B^T B)^{-1} B^T g_{lora}^B (A^T A)^{-1}. \tag{58}$$

The comparison of the selection of $X$ is presented in Table 6. As shown in the table, the choice of $X$ significantly impacts LoRA-Pro's performance, particularly evident in the GSM8K dataset. Different $X$ selections influence the subspaces of $A$ and $B$, ultimately affecting the optimization process described in Theorem 2.1. Our experiments demonstrate that the Sylvester solution consistently outperforms both the zero and symmetry solutions across all three evaluation tasks. We attribute the superior performance of the Sylvester solution to its ability to select subspaces for $A$ and $B$ that enable faster gradient descent (i.e., maximizing the approximation between the modified gradients $g^A, g^B$ and the LoRA gradients $g_{lora}^A, g_{lora}^B$g).

Table 6: Ablation study on the selection of different X in LoRA-Pro.

| Choice of X | MT-Bench | GSM8K | HumanEval |
|---|---|---|---|
| Zero | 5.58±0.14 | 31.74±0.69 | 17.28±0.35 |
| Symmetry (Eq. (58)) | 5.71±0.11 | 42.81±0.62 | 17.88±0.35 |
| Sylvester (Thm. 2.3) | 5.72±0.03 | 57.57±0.50 | 22.97±0.35 |

## D.2    VISUALIZATION OF DIFFERENCES BETWEEN EQUIVALENT GRADIENTS AND FULL GRADIENTS

In this section, we fine-tune Llama-2-7B on the MetaMathQA100k dataset and visualize the discrepancies between the equivalent gradients of LoRA and LoRA-Pro and the full gradients during training, i.e., the differences before and after gradient adjustments. We present visualizations for different optimization modules, including the Q, K, V, O, Up, Down, and Gate layers, and provide results for these modules across the shallow (1), medium (15), and deep (31) layers of Llama-2-7B.

The results are shown in Figure 4. From the figure, we can draw the following conclusions:

- After gradient adjustments in LoRA-Pro, we observe a significant reduction in the distance between the equivalent gradients and the full gradients.

- In certain layers, the discrepancy between LoRA's equivalent gradients and the full gradients continues to increase (e.g., Layer 1 O, Up, Gate projections; Layer 15 Up and Gate projections; and Layer 31 O projection). However, in these layers, the discrepancy for LoRA-Pro remains stable, indicating that LoRA-Pro can consistently align with the full gradients during training, preventing the model from settling into sub-optimal solutions.

- In deep layers, the discrepancy between equivalent gradients and full gradients decreases as training progresses, whereas in shallow and medium layers, the discrepancy first increases and then stabilizes. The cause of this phenomenon is not yet clear, and we plan to investigate it further in future research.

These findings highlight that LoRA-Pro effectively reduces the distance between LoRA and full gradients during training and ensures continuous alignment with full gradients, underscoring the efficacy of LoRA-Pro.

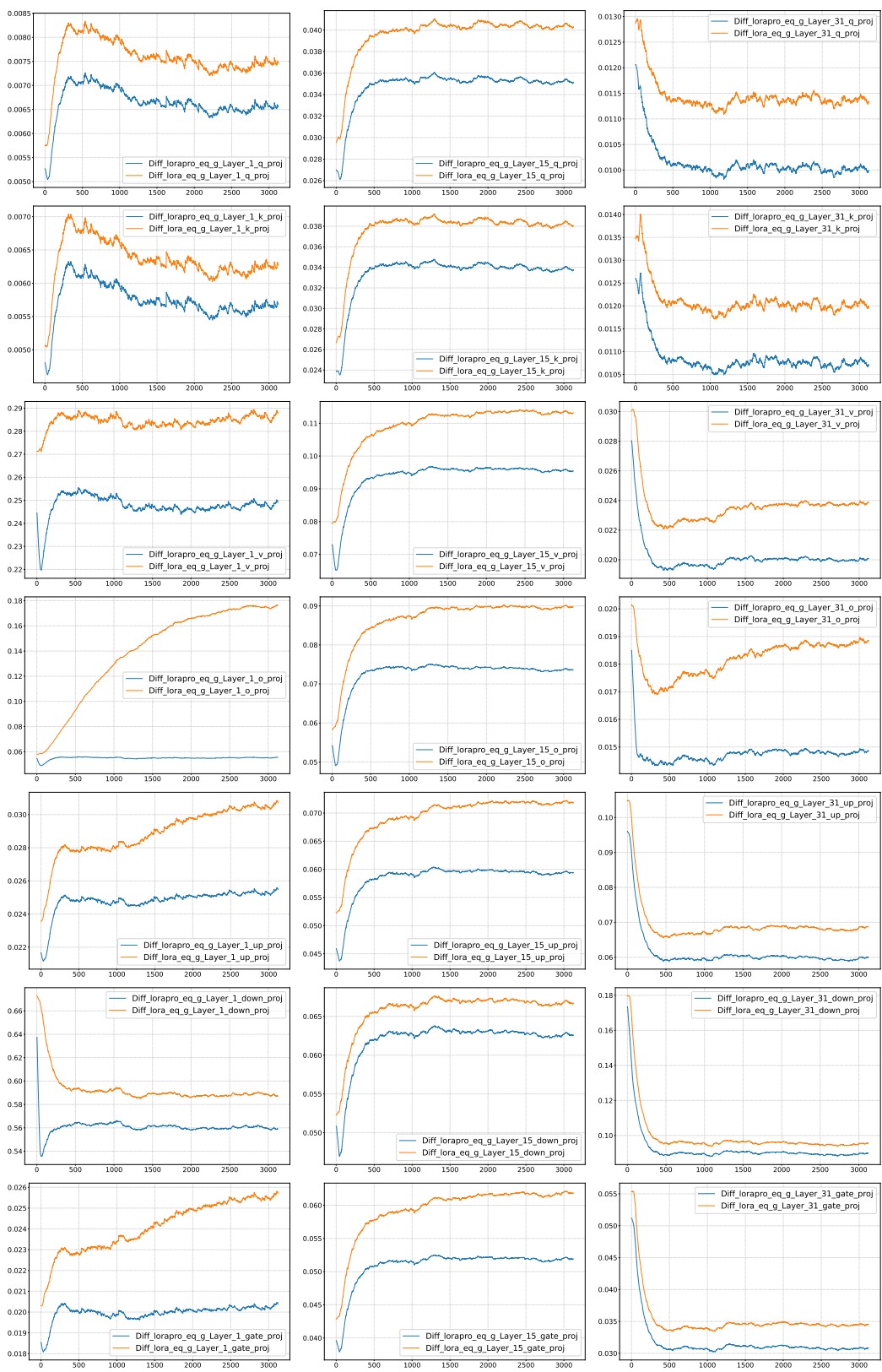

Figure 4: Visualization of the differences between the equivalent gradients of LoRA, LoRA-Pro, and the full-parameter gradients during training, i.e., $\|\tilde{g} - g\|_F$. The rows illustrate the differences across various modules, including Q, K, V, O, Up, Down, and Gate. The columns show the differences at different depths, categorized as shallow (1), medium (15), and deep layers (31).

### D.3 EXPERIMENTS RESULTS WITH DIFFERENT LEARNING RATES

To demonstrate the effectiveness of LoRA-Pro, we evaluated its performance on GSM8K under learning rates of 1e-5 and 5e-5, comparing it with LoRA and LoRA-GA. The results, presented in Table 7, show that LoRA-Pro maintains its advantages under both learning rates, highlighting its robustness to variations in learning rate.

Table 7: Performance comparison of LoRA, LoRA-GA, LoRA-Pro on GSM8K with learning rates 1e-5, 2e-5, and 5e-5.

| GSM8K | LoRA | LoRA-GA | LoRA-Pro |
|-------|------|---------|----------|
| 1e-5 | 36.65±0.82 | 50.25±0.62 | 56.48±0.27 |
| 2e-5 | 42.08±0.04 | 53.60±0.30 | 57.57±0.50 |
| 5e-5 | 46.41±0.16 | 52.89±0.19 | 58.76±1.86 |

### D.4 ADDITIONAL EXPERIMENTS ON LATEST MODELS

Table 8: Performance comparison of LoRA, LoRA-GA, and LoRA-Pro with Llama-2-7B and Llama-3.1-8B.

| GSM8K | LoRA | LoRA-GA | LoRA-Pro |
|-------|------|---------|----------|
| Llama-2-7B | 42.08±0.04 | 53.60±0.30 | 54.23±0.79 |
| Llama-3.1-8B | 71.04±0.26 | 72.20±1.15 | 75.49±0.42 |

To further demonstrate the effectiveness of LoRA-Pro, we conducted additional experiments using the latest model, LLaMA-3.1-8B (Dubey et al., 2024). We fine-tuned the model using these three methods,LoRA, LoRA-GA, and LoRA-Pro,on the MetaMath100k dataset and evaluated its performance on the GSM8k dataset. All results are averaged over three different random seeds.

As shown in Table 8, LoRA-Pro demonstrates a clear advantage over both LoRA and LoRA-GA when applied to the LLaMA-3.1-8B model, further highlighting its effectiveness.

