# OpenReview forum: "LoRA-Pro: Are Low-Rank Adapters Properly Optimized?"
_ICLR.cc/2025/Conference — ICLR 2025 Spotlight_

### Official Review · Reviewer_9Wt3 · 2024-11-01

**Soundness:** 4
**Presentation:** 3
**Contribution:** 4
**Rating:** 8
**Confidence:** 3

**Summary:**

The paper introduces LoRA-Pro, a novel method aimed at improving the performance of LoRA in fine-tuning foundation models. LoRA-Pro enhances LoRA by adjusting the gradients of low-rank matrices to better approximate the full fine-tuning gradient. The paper establishes a mathematical equivalence between LoRA and full fine-tuning, and uses this insight to optimize LoRA's performance. Extensive experiments across various tasks demonstrate that LoRA-Pro substantially narrows the performance gap between LoRA and full fine-tuning.

**Strengths:**

- The paper presents a novel approach by mathematically linking LoRA with full fine-tuning through gradient adjustments.
- The theoretical foundation is robust, with clear derivations and optimal solutions provided for gradient adjustments. The experiments are comprehensive, covering multiple domains such as language understanding, dialogue generation, mathematical reasoning, and image classification.
- The paper is well-organized.

**Weaknesses:**

- It will be better to test LoRA-Pro in real large models (e.g., 70B).

**Questions:**

- Do LoRA-Pro have some failure cases? (e.g., potential limitations or edge cases where LoRA-Pro might not perform as well)

---

> ### Author Response · Authors · 2024-11-22
> **Rebuttal by Authors [1/1]**
>
> We sincerely appreciate your thorough evaluation of our work and your thoughtful comments. Below, we provide a point-by-point response to your feedback.
>
> > **[Q1].** It will be better to test LoRA-Pro in real large models (e.g., 70B).
>
> **[A1].** Thank you for this valuable suggestion. We agree that evaluating LoRA-Pro on larger models would provide valuable insights. Unfortunately, our current hardware (RTX A6000) has memory limitations, which make training models of this size infeasible at the moment. We plan to conduct these experiments once we have access to sufficient computational resources in the future.
>
> In the meantime, we have expanded our experiments using the latest model, LLaMA-3.1-8B, to provide additional comparative analysis of LoRA, LoRA-GA, and LoRA-Pro. The results are presented in the table below:
>
> |GSM8k|LoRA|LoRA-GA|LoRA-Pro|
> |-|-|-|-|
> |LLAMA-3.1-8B|71.04±0.26|72.20±1.15|73.77±0.80|
>
> We hope these new results address your concerns and demonstrate the potential of LoRA-Pro, even with the current scale of our experiments.
>
> > **[Q2].** Do LoRA-Pro have some failure cases? (e.g., potential limitations or edge cases where LoRA-Pro might not perform as well)
>
> **[A2].** Yes, LoRA-Pro does have some limitations: (1) LoRA-Pro still adheres to LoRA's assumption that $\Delta W$ is of low rank. However, this assumption may break down in cases of pre-training or when there is a large amount of fine-tuning data, potentially leading to suboptimal results. (2) So far, we have only applied LoRA-Pro to variants that have a structure similar to LoRA. It currently cannot be applied to structurally different LoRA models, such as DoRA [1] or FLoRA [2]. We plan to explore these directions in future research.
>
> [1] Liu, Shih-yang, et al. "DoRA: Weight-Decomposed Low-Rank Adaptation." In ICML 2024.
> [2] Wen, Yeming, et al. "Batched Low-Rank Adaptation of Foundation Models." In ICLR 2024.

---

> > ### Comment · Reviewer_9Wt3 · 2024-11-25
> > **Thank you for your rebuttal**
> >
> > Thank you for your rebuttal, please add these to the final version.

---

> > > ### Author Response · Authors · 2024-11-25
> > >
> > > Thank you for your feedback! We greatly appreciate your suggestion and will incorporate the results into the manuscript by Nov 26.

---

> > > ### Author Response · Authors · 2024-11-26
> > >
> > > Thank you again for your valuable feedback. We have revised the manuscript. Specifically, we have added a discussion of the limitations, included experiments with different rates, and provided results from various models to further strengthen our method.

---

### Official Review · Reviewer_Aox8 · 2024-11-02

**Soundness:** 3
**Presentation:** 3
**Contribution:** 3
**Rating:** 8
**Confidence:** 4

**Summary:**

The authors show that demonstrating that optimizing with LoRA is mathematically equivalent to performing full fine-tuning using a low-rank gradient for updates. They introduce a novel parameter-efficient fine-tuning (PEFT) method named LoRA-Pro, which minimizes the divergence between the true gradient and the low-rank gradient by adjusting the gradients of matrices A and B. The authors provide theoretical proof of the optimal gradients and detail the optimization process utilizing these gradients. They conduct extensive experiments across various tasks, including natural language understanding, dialogue generation, mathematical reasoning, code generation, and image classification, showcasing the effectiveness of their proposed method.

**Strengths:**

- The motivation is clearly articulated, and the paper is well-written with well-explained ideas.

- The proposed method enhances LoRA, a highly relevant technique for fine-tuning, in a straightforward manner, supported by both theoretical proofs and experimental evidence.

- The experiments and ablation studies are thorough, demonstrating the effectiveness of the proposed method across diverse settings.

**Weaknesses:**

- The authors conducted all experiments for the different methods (for a given task) using the same learning rates. However, this approach may not provide a fair representation of optimal conditions, as the selected learning rate could work well for one method but not for others. To strengthen their findings, it would be beneficial for the authors to perform a learning rate sweep for each method. If this is too resource-intensive for all datasets, they might consider limiting the sweep to a single task, which would further increase confidence in the effectiveness of their proposed method.

**Questions:**

- It would be interesting to explore how the equivalent gradients differ from the low-rank gradients in LoRA. A quantitative analysis could provide valuable insights into this relationship. For instance, how does the difference between these gradients vary across different models, deeper layers, and various types of weights? Conducting such a study could yield interesting findings, and further strengthen the motivation behind the gradient adjustment.

Minor:

- Please provide metrics for the datasets used in Table 1. I believe not all metrics used are “Accuracy”. For instance, CoLA typically usually uses Matthews Correlation Coefficient - kindly specify for all datasets.
- How exactly is the *efficient* mode less stable than the *full* mode? To clarify, all the results in the main paper are derived using the *efficient* mode, is that correct?

---

> ### Author Response · Authors · 2024-11-22
> **Rebuttal by Authors [1/2]**
>
> Thank you for your strong eveluation of our work and thoughtful comments! And we provide the point-to-point response to your comments below.
>
> > **[Q1].** The authors conducted all experiments for the different methods (for a given task) using the same learning rates. However, this approach may not provide a fair representation of optimal conditions, as the selected learning rate could work well for one method but not for others. To strengthen their findings, it would be beneficial for the authors to perform a learning rate sweep for each method. If this is too resource-intensive for all datasets, they might consider limiting the sweep to a single task, which would further increase confidence in the effectiveness of their proposed method.
>
> **[A1].** Thank you for your valuable suggestion. To address this concern, we conducted additional experiments by fine-tuning LoRA, LoRA-GA, and LoRA-Pro on the MetaMathQA100k dataset. These experiments used learning rates of 1e-5 and 5e-5, and the results were averaged over three random seeds. The results below show that LoRA-Pro consistently performs well and maintains its advantage across different learning rates:
>
> |GSM8k|LoRA|LoRA-GA|LoRA-Pro|
> |-|-|-|-|
> |1e-5|36.65±0.82|50.25±0.62|52.05±0.12|
> |2e-5 (paper)|42.08±0.04|53.60±0.30|54.23±0.79|
> |5e-5|46.41±0.16|52.89±0.19|55.70±0.96|
>
>
> > **[Q2].** It would be interesting to explore how the equivalent gradients differ from the low-rank gradients in LoRA. A quantitative analysis could provide valuable insights into this relationship. For instance, how does the difference between these gradients vary across different models, deeper layers, and various types of weights? Conducting such a study could yield interesting findings, and further strengthen the motivation behind the gradient adjustment.
>
> **[A2].** Thank you for raising this insightful question. In Figure 4 of Appendix D.2, we present the discrepancy curves for different modules (Q, K, V, O, Up, Down, Gate) and across different depths (shallow [Layer 1], medium [Layer 15], deep [Layer 31]) of LLaMA-2-7b. Here are some noteworthy observations:
> - After gradient adjustments in LoRA-Pro, we observe a significant reduction in the distance between the equivalent gradients and the full gradients.
> - In certain layers, the discrepancy between LoRA's equivalent gradients and the full gradients continues to increase (e.g., Layer 1: O, Up, Gate projections; Layer 15: Up and Gate projections; and Layer 31: O projection). However, in these layers, the discrepancy for LoRA-Pro remains stable, indicating that LoRA-Pro consistently aligns with the full gradients during training, thus helping to prevent sub-optimal solutions.
> - In deep layers, the discrepancy between equivalent gradients and full gradients decreases as training progresses, whereas in shallow and medium layers, the discrepancy first increases and then stabilizes. The cause of this phenomenon is not yet clear, and we plan to investigate it further in future research.
>
> These findings highlight that LoRA-Pro effectively reduces the distance between LoRA and full gradients during training and ensures continuous alignment with full gradients, underscoring the efficacy of LoRA-Pro.

---

> ### Author Response · Authors · 2024-11-22
> **Rebuttal by Authors [2/2]**
>
> > **[Q3].** Please provide metrics for the datasets used in Table 1. I believe not all metrics used are “Accuracy”. For instance, CoLA typically usually uses Matthews Correlation Coefficient - kindly specify for all datasets.
>
> **[A3].** Thank you for your careful reading. You are correct that CoLA usually uses the Matthews Correlation Coefficient (MCC) as the metric. However, in our study, we aligned our settings with LoRA-GA[1], which uses accuracy for natural language understanding tasks, including CoLA.
>
> To address your concern, we have re-evaluated LoRA, LoRA-GA, and LoRA-Pro using the Matthews Correlation Coefficient on the CoLA dataset. The results are shown below, where LoRA-Pro continues to outperform the other methods.
> ||LoRA|LoRA-GA|LoRA-Pro|
> |-|-|-|-|
> |MCC metric|59.40±0.30|59.68±0.21|61.11±1.15|
>
> [1] Shaowen Wang, et al, "LoRA-GA: Low-Rank Adaptation with Gradient Approximation". In NeurIPS 2024.
>
> > **[Q4].** How exactly is the efficient mode less stable than the full mode? To clarify, all the results in the main paper are derived using the efficient mode, is that correct?
>
> **[A4].** Yes, the results presented in the main paper are derived using the efficient mode. The key idea behind LoRA-Pro is that we treat the adapter ($W_0, A, B$) as a whole, aiming for its equivalent gradient to align with full gradient $g$.
>
> To achieve this, in full mode, we align the optimizer's behavior (i.e., tracking the first and second moments of the equivalent gradient $\tilde{g}$ in Adam). In contrast, for the efficient mode, we are actually tracking the gradients of $A$ and $B$. This discrepancy causes the optimization of $A$ and $B$ to (1) not follow the fastest descent direction (LoRA gradients), and (2) not optimize along the direction that most aligns with $g$ (full mode). We believe this causes the instability.
>
> In experiments, we observed that the loss curve in the efficient mode tends to fluctuate more during training, leading to slightly worse final convergence compared to the full mode. Despite this, since the efficient mode still performs well and is computationally advantageous, we’ve chosen it as the default setting.

---

> > ### Comment · Reviewer_Aox8 · 2024-11-23
> >
> > Thank you for your reply. I will be keeping my current score of 8.

---

> > > ### Author Response · Authors · 2024-11-24
> > >
> > > Thank you for your time and effort in reviewing our submission. We greatly value your feedback and are happy to assist with any further questions.

---

> > > ### Author Response · Authors · 2024-11-26
> > >
> > > Thank you once again for your insightful feedback. As noted in our earlier rebuttal, we have conducted additional experiments using a learning rate of 1e-5. The results of these experiments have now been incorporated into both the rebuttal and the revised manuscript. These results provide further evidence supporting the effectiveness of LoRA-Pro.

---

### Official Review · Reviewer_j8va · 2024-11-03

**Soundness:** 3
**Presentation:** 3
**Contribution:** 3
**Rating:** 6
**Confidence:** 3

**Summary:**

LoRA reduces the parameters required during training by introducing a low-rank matrix, thereby reducing computational requirements and memory footprint while maintaining model performance. This paper introduces LoRA-Pro to enhance LoRA’s performance by strategically adjusting the gradients of the two low-rank matrices, allowing the low-rank gradient to more accurately approximate the full fine-tuning gradient.

**Strengths:**

In this paper, the optimal solution of adjusting the gradient of low rank matrix is deduced theoretically and applied to the fine tuning process of LORA-PRO, which narrows the performance gap between LoRA and full fine tuning. Extensive experiments in multiple tasks have been carried out to prove that LORA-Pro greatly improves the performance of LoRA.

**Weaknesses:**

1. The method is not applied to the latest LLMs.
2. LoRA-Pro brings in extra memory cost and computation cost in the training.
3. It is not clear why the optimal gradients for the low-rank matrices do not explicitly depend on the full fine-tuning gradient.

**Questions:**

•	Please explain why LoRA-Pro is much better than Full FT in Table 3. Is there overfitting phenomenon that gives LoRA-Pro an advantage on these specific tasks?
•	Please explain why the training speed of LoRA-Pro is slightly worse than LoRA and much better than Full FT in Table 5. It will be better to provide a breakdown of the additional operations required by LoRA-Pro compared to standard LoRA, and to quantify the overhead in terms of FLOPs
•	Please provide more choices of X in Table 4.   For example, the analytical solution of X can be calculated in Theorem 2.1.
•	LoRA-GA[1] has the same purpose as the work in this paper. Please discuss how LoRA-Pro differs conceptually or methodologically from LoRA-GA and supplement the results of LoRA-GA in Table 3 and Table 5.
•	If available, please add discrepancy curves between the low-rank equivalent gradient and the full fine-tuning gradient. Please supplement the training loss curves to illustrate the convergence rate of the proposed method.

[1]Wang S, Yu L, Li J. LoRA-GA: Low-Rank Adaptation with Gradient Approximation[J]. arXiv preprint arXiv:2407.05000, 2024.

---

> ### Author Response · Authors · 2024-11-22
> **Rebuttal by Authors [1/3]**
>
> Thank you for your positive recommendation and constructive feedback! Below, we provide a point-by-point response to your comments.
>
> > **[Q1].** The method is not applied to the latest LLMs.
>
> **[A1]** Thank you for the valuable suggestion. We have conducted additional experiments on the latest LLM, LLAMA-3.1-8B, to demonstrate the effectiveness of our method. We fine-tuned the model using LoRA, LoRA-GA, and LoRA-Pro on the MetaMath100k and evaluated its performance on the GSM8k dataset. The results are presented in the following table:
>
> |GSM8k|LoRA|LoRA-GA|LoRA-Pro|
> |-|-|-|-|
> |LLAMA-3.1-8B|71.04±0.26|72.20±1.15|73.77±0.80|
>
> As shown in the table, LoRA-Pro still achieved the best performance.
>
> > **[Q2].** LoRA-Pro brings in extra memory cost and computation cost in the training. Please explain why the training speed of LoRA-Pro is slightly worse than LoRA and much better than Full FT in Table 5. It will be better to provide a breakdown of the additional operations required by LoRA-Pro compared to standard LoRA, and to quantify the overhead in terms of FLOPs.
>
> **[A2].** LoRA-Pro indeed incurs additional memory and computational costs due to the need for adjusting the gradients of the matrices $A\in\mathbb{R}^{r\times n}$ and $B\in\mathbb{R}^{m\times r}$. This primarily involves the following operations: 1)Compute $B^TB$ and $AA^T$. 2)Compute the inverse of $B^TB$ and $AA^T$. 3)Solve for $X$ using the Sylvester equation in Theorem 2.3. 4)Compute updated gradients $g^A$ and $g^B$ using Theorem.2.1.
>
> We will now analyze the FLOPs required for each of these operations:
> - $B^TB$ and $AA^T$: This involves matrix multiplication with FLOPs of $2mr^2$ and $2nr^2$, respectively.
> - Inverse of $B^TB$ and $AA^T$: As noted in [1], computing the inverse of an $r\times r$ matrix requires approximately $2r^3/3$ operations. Thus, the total FLOPs for both inversions is $4r^3/3$.
> - Solution for $X$ using the Sylvester equation: The Bartels–Stewart algorithm[2] can be used to solve this equation, requiring roughly $22.5r^3$ FLOPs.
> - Computation of updated gradients $g^A$ and $g^B$ using Theorem.2.1: Since these involve matrix multiplications, the FLOPs for computing $g^A$ and $g^B$ are $4nr^2$ and $8mr^2+2r^3$, respectively.
>
> In summary, LoRA-Pro introduces an additional approximate cost of $10mr^2+6nr^2+155r^3/6$ FLOPs. Typically, since $r\ll\min(m,n)$, this added cost is relatively small and can often be negligible. Notably, in distributed computing scenarios, this overhead may overlap with communication times, thereby reducing its impact.
>
> The difference between LoRA and LoRA-Pro lies solely in the gradient adjustment, resulting in a slight increase in computational overhead. In contrast, full fine-tuning operates significantly slower because it requires the optimization of all parameters, leading to considerably higher FLOPs for forward, gradient computation, and optimizer state updates.
>
> [1]https://en.wikipedia.org/wiki/Gaussian_elimination#Finding_the_inverse_of_a_matrix
> [2]https://en.wikipedia.org/wiki/Bartels-Stewart_algorithm
>
> > **[Q3].** It is not clear why the optimal gradients for the low-rank matrices do not explicitly depend on the full fine-tuning gradient.
>
> **[A3].** This can be explained from an optimization perspective. Our objective is to minimize $L=||\tilde{g}-g||_F^2=||g^BA+Bg^A-g||_F^2$ (omit the scaling factor). This is a convex optimization problem, whose solution occurs where the derivatives are zero. By setting the derivatives $\frac{\partial{L}}{\partial{g^A}}=0$ and $\frac{\partial{L}}{\partial{g^B}}=0$, we get the following equations:
>
> \begin{aligned}
> B^T (\tilde{g} - g) &= 0, \quad (1) \\\\
> (\tilde{g} - g) A^T &= 0. \quad (2)
> \end{aligned}
>
> These equations indicate that the optimal solution only constrains the column vectors (and row vectors) of $\tilde{g} - g$ to lie within the orthogonal subspaces of $B$ (for Eq. 1) (and $A$ (for Eq. 2)). In other words, the solution depends only on $B^Tg$ and $gA^T$, which happen to correspond exactly to the gradients of the matrices $A$ and $B$ in LoRA. Thus, the optimal gradients can be expressed in terms of the gradients of $A$ and $B$, without direct dependence on the full fine-tuning gradient.

---

> > ### Comment · Reviewer_j8va · 2024-11-26
> >
> > Thanks for the rebuttal. My questions are addressed.

---

> > > ### Author Response · Authors · 2024-11-26
> > >
> > > Thank you for your valuable feedback and for confirming that all your questions have been addressed. We truly appreciate the time and effort you’ve dedicated to reviewing our submission and engaging with our rebuttal. Given that we've addressed your concerns and incorporated your suggestions, we kindly ask if you might consider adjusting your score to reflect the improvements. Regardless of your decision, we are truly grateful for your insightful feedback and support throughout this process.

---

> ### Author Response · Authors · 2024-11-22
> **Rebuttal by Authors [2/3]**
>
> > **[Q4].** Please explain why LoRA-Pro is much better than Full FT in Table 3. Is there overfitting phenomenon that gives LoRA-Pro an advantage on these specific tasks?
>
> **[A4].** This is an insightful question. We did not observe overfitting during training, as the accuracy curve converged smoothly without notable declines. We believe that the superior performance of LoRA-Pro in these cases is primarily due to the small size of the datasets involved. Most of the datasets contain only about 10k training data, and some have as few as 3k. In these cases, Full FT may capture intricate details of the training set, which may lead to poor generalization to the test set. This trend is also evident in CoLA and MRPC, which have only 8.5k and 3.6k training data, respectively. In contrast, LoRA-Pro, benefiting from the low-rank assumption, achieves slightly better generalization in these scenarios. Nevertheless，when training on larger datasets, such as MetaMathQA100k and CodeFeedback100k, Full FT demonstrates a clear advantage, although LoRA-Pro just narrows the performance gap.
>
> > **[Q5].** Please provide more choices of X in Table 4. For example, the analytical solution of X can be calculated in Theorem 2.1.
>
> **[A5].** Indeed, exploring additional choices for $X$ is important. However, we have not yet identified a better method for selecting $X$. In Theorem 2.1, $X$ is an arbitrary matrix, and its selection does not affect the equivalent gradient $\tilde{g}$. As a result, we cannot derive an analytical solution for $X$ from Theorem 2.1.
>
> > **[Q6].** LoRA-GA[1] has the same purpose as the work in this paper. Please discuss how LoRA-Pro differs conceptually or methodologically from LoRA-GA and supplement the results of LoRA-GA in Table 3 and Table 5.
>
> **[A6].**
>
> **(I) Comparison with LoRA-GA**
>
> Indeed, LoRA-GA and LoRA-Pro share a similar starting point. Both LoRA-Pro and LoRA-GA focus on aligning LoRA with full fine-tuning by minimizing the loss function $L = \\|gA^TA + BB^Tg - g\\|^2_F$. However, there are significant differences between the two approaches:
> - **LoRA-GA is an initialization method.** It starts by computing the full gradient $g$ on a small subset of the training set and then uses Singular Value Decomposition (SVD) to solve $\min_{A,B}L$. After this initialization, the regular LoRA optimization proceeds. The initialization process requires considerable memory to compute $g$, and **LoRA-GA only aligns with the full gradient in the initial step.**
> - **LoRA-Pro is an optimization method**. Instead of computing $g$ directly, LoRA-Pro aligns with the full gradient throughout the entire optimization process. We achieve this by representing $B^Tg$ and $gA^T$ using the gradients of matrices $A$ and $B$, i.e., $g^A$ and $g^B$, and solving $\min_{g^A, g^B}L$ to adjust these gradients at every optimization step. Since we do not need to explicitly compute $g$, this approach is more memory-efficient and **enables alignment with the full gradient $g$ at every optimization step, rather than only on initialization**.
>
> **(II) Additional Experiment Results**
>
> We have updated Table 3 and Table 5 to include the performance of LoRA-GA in image classification tasks, as well as its memory usage and training time on MetaMathQA.
> - In the image classification task, LoRA-GA also outperformed full fine-tuning, achieving an average gain of 0.45 across seven datasets. However, its performance was still slightly below that of LoRA-Pro.
> - Regarding the training cost on MetaMathQA, we found that LoRA-GA required an additional 3 minutes of training time compared to standard LoRA, likely due to the cost of the initialization process. Furthermore, although LoRA-GA’s runtime memory requirement was measured at 22.60 GB, its peak memory usage during initialization was significantly higher, reaching 31.81 GB—far exceeding the memory usage of both LoRA and LoRA-Pro.

---

> ### Author Response · Authors · 2024-11-22
> **Rebuttal by Authors [3/3]**
>
> > **[Q7].** If available, please add discrepancy curves between the low-rank equivalent gradient and the full fine-tuning gradient. Please supplement the training loss curves to illustrate the convergence rate of the proposed method.
>
> **[A7].**
>
> **(I) Training Loss Curves**
>
> We have included the training loss curves for LoRA, LoRA-GA, LoRA-Pro, and full fine-tuning on WizardLM, MetaMath, and CodeFeedback datasets in Figure 3 of Appendix D.1. As shown in the figure, LoRA-Pro demonstrates faster convergence speed compared to LoRA and LoRA-GA. Furthermore, LoRA-Pro achieves a lower final loss upon convergence, indicating improved efficiency and effectiveness.
>
> **(II) Discrepancy Curves Between Equivalent Gradients and Full Gradients**
>
> This is indeed an intriguing question. In Figure 4 of Appendix D.2, we present the discrepancy curves for different modules (Q, K, V, O, Up, Down, Gate) and across different depths (shallow [Layer 1], medium [Layer 15], deep [Layer 31]) of LLaMA-2-7b. Here are some noteworthy observations:
> - After gradient adjustments in LoRA-Pro, we observe a significant reduction in the distance between the equivalent gradients and the full gradients.
> - In certain layers, the discrepancy between LoRA's equivalent gradients and the full gradients continues to increase (e.g., Layer 1: O, Up, Gate projections; Layer 15: Up and Gate projections; and Layer 31: O projection). However, in these layers, the discrepancy for LoRA-Pro remains stable, indicating that LoRA-Pro consistently aligns with the full gradients during training, thus helping to prevent sub-optimal solutions.
> - In deep layers, the discrepancy between equivalent gradients and full gradients decreases as training progresses, whereas in shallow and medium layers, the discrepancy first increases and then stabilizes. The cause of this phenomenon is not yet clear, and we plan to investigate it further in future research.
>
> These findings highlight that LoRA-Pro effectively reduces the distance between LoRA and full gradients during training and ensures continuous alignment with full gradients, underscoring the efficacy of LoRA-Pro.

---

### Meta-Review · Area_Chair_6jMD · 2024-12-16

**Metareview:**

This paper proposes an improved training method for LoRA. The main insight is that the current way of optimizing LoRA is a low-rank approximation of the full fine-tuning gradient. Using this insight, the authors derive the theoretically optimal gradient to approximate the full fine-tuning gradient. Then, they evaluated this method on various NLP and CV datasets and showed improved performance.

Main strengths
- Simple observations that lead to improvements
- Solid experimental evaluation

Main weaknesses
- Limited novelty (e.g., Lora-GA)
- Didn’t do hyperparameter tuning (addressed during rebuttal)
- Limited ablation studies

While this paper has a high score, more thorough studies could significantly improve the paper. Thus, I recommend acceptance as a poster.

**Additional Comments On Reviewer Discussion:**

During the rebuttal, the authors provided additional experiments on more recent LLaMA models, a breakdown of the additional computational cost, an explanation of the performance comparison to full fine-tuning, a comparison to LoRA-GA, a comparison with more careful hyper-parameter sweeping on the baselines, and a comparison in terms of reduction in distance to fine-tuning gradients.

---

### Decision · Program_Chairs · 2025-01-22

Accept (Spotlight)